# Correlation between genetic and environmental risk factors for age-related macular degeneration in Brazilian patients

Priscila H. H. Rim[1]*, José Paulo C. de Vasconcellos[1], Mônica B. de Melo[2], Flavio M. C. Medina[3], Daniela P. D. Sacconi[2], Tamires P. Lana[2], Fabio E. Hirata[1], Luis A. Magna[4], Antonia P. Marques-de-Faria[4]

1 Department of Ophthalmology, School of Medical Sciences, University of Campinas (UNICAMP), Campinas, São Paulo, Brazil, 2 Laboratory of Human Genetics, Center for Molecular Biology and Genetic Engineering (CBMEG), University of Campinas (UNICAMP), Campinas, São Paulo, Brazil, 3 Department of Ophthalmology, School of Medical Sciences, Rio de Janeiro State University (UERJ), Rio de Janeiro, Rio de Janeiro, Brazil, 4 Department of Medical Genetics, School of Medical Sciences, University of Campinas (UNICAMP), Campinas, São Paulo, Brazil

* priscilarim@gmail.com

**Data Availability Statement:** All relevant data are within the paper and its Supporting information files.

## Abstract

### Purpose

To analyze the correlations between age-related macular degeneration (AMD) and genetic and environmental risk factors for in a Brazilian population.

### Design

Cross-sectional study with a control group.

### Methods

We collected data on 236 participants 50 years of age or older (141 with AMD and 95 controls without the disease). Data was obtained using a questionnaire and included information on demographics, ocular and medical history, family history of AMD, lifestyle, and smoking and drinking habits. Genetic evaluations included direct sequencing for the *LOC387715 (rs10490924)* variant, as well as PCR and enzymatic digestion for the *CFH Y402H (rs1061170)* and *HTRA1 (rs11200638)* variants. We performed a risk assessment of environmental risk factors and genetic variants associated with AMD and determined correlations between AMD and the data collected using multiple linear regression analysis.

### Results

Of the 141 AMD cases, 99 (70%) had advanced AMD in at least one eye (57% neovascular AMD and 13% geographic atrophy), and 42 (30%) had not-advanced AMD. Family history of AMD (OR: 6.58; 95% CI: 1.94–22.31), presence of cardiovascular disease (CVD) (OR: 2.39; 95% CI: 1.08–5.28), low physical activity level (OR: 1.39; 95% CI: 0.82–2.37), and high serum cholesterol (OR: 1.49; 95% CI: 0.84–2.65) were associated with an increased risk for AMD. There was a significant association between CVD and incidence of advanced

**Funding:** This study was supported by National Council of Technological and Scientific Development (CNPq) [Grant #472645/2008] and by São Paulo Research Foundation (FAPESP) [grant 2010/18353–9].

**Competing interests:** The authors have declared that no competing interests exist.

AMD (OR: 2.29; 95% CI 0.81–6.44). The OR for the risk allele of the *LOC387715* gene, the *CFH* gene and the *HTRA1* gene were 2.21 (95% CI: 1.47–3.35), 2.27 (95% CI: 1.52–3.37), and 2.76 (95% CI: 1.89–4.03), respectively. In the stepwise multiple linear regression analyses, the *HTRA1* and *CFH* risk alleles, family history of AMD, the *LOC387715* risk allele, and CVD were associated with an increased risk of AMD for a total of 25.6% contribution to the AMD phenotype.

## Conclusions

The analysis correlating environmental and genetic risk factors such as family history of AMD, and CVD and the variants of *HTRA1*, *CFH*, and *LOC387715* genes showed an expressive contribution for the development of AMD among this admixed population.

## Introduction

Age-related macular degeneration (AMD; MIM 603075) is characterized by progressive degeneration of the photoreceptors and retinal pigment epithelial complex, primarily in the macular region of the retina, resulting in irreversible central vision loss. According to the WHO, AMD (along with cataract and glaucoma) is one of the major causes of blindness in elderly populations [1, 2]. It is considered the leading cause of irreversible blindness and visual impairment among individuals older than 50 years of age in developed countries, affecting up to 25% of individuals 75 years of age or older. The disease burden is likely to grow, since the number of people affected worldwide is expected to double by 2050 [3].

AMD is a complex condition that involves genetic and environmental factors. The major risk factors for AMD include genetics, demography, nutrition, lifestyle, other environmental factors, and ocular factors. As confirmed in numerous population studies, age represents the strongest risk factor for AMD [4, 5].

Family history has also been found to be a major risk factor [6], leading several investigators to search for the genetic basis of this condition. Genome-wide association studies have identified several variants of genes associated with AMD, the majority of which are present in populations of European ancestry [7]. The complement factor H (*CFH*) single nucleotide polymorphism (SNP) *rs1061170* on chromosome 1q31 was the first variant found to be associated with AMD [8]. Later, a susceptibility locus for AMD risk was identified on chromosome 10q26, and two variants were identified in this region [9]. The SNP *rs10490924* lies within the *LOC387715/ARMS2* gene, now known as age-related maculopathy susceptibility 2 (*ARMS2*), which has been implicated as a strong genetic risk factor for AMD [10, 11]. The other SNP, variant *rs11200638*, lies within the promoter region of the gene known as high temperature requirement factor A1 (*HTRA1*), located approximately 7 kb downstream from *LOC387715/ARMS2*, and it has been associated with late AMD [12]. The presence of these variants constitutes a major risk of disease and accounts for approximately 30% to 60% of the genetic risk of developing AMD in populations of European descendant [13].

Environmental factors have been associated with a higher incidence and progression of AMD, and smoking intensity is the most consistent environmental risk factor for developing advanced forms of the disease [4, 14]. According to some authors, AMD and cardiovascular disease (CVD) may share a similar pathogenesis, as well as similar risk factors, including a smoking habit, elevated total serum cholesterol, higher body mass index (BMI), lipid

metabolism, and hypertension [15–17]. The prevalence of both conditions is strongly associated with increasing age and vascular abnormalities caused by atherosclerosis. A report on cardiovascular risk factors in a Brazilian population 60 years of age and older found 87% of subjects to have at least two risk factors from a list that included history of hypertension, a smoking habit, dyslipidemia, and a sedentary lifestyle [18]. Another factor related to CVD is physical activity: some studies have suggested its possible role in improving cardiovascular conditions and lowering the risk of AMD development [17, 19, 20]. Among the ocular risk factors, the risk of developing late AMD is higher in eyes that have undergone cataract surgery, according to population-based studies such as the longitudinal Beaver Dam Eye Study and the Blue Mountains Eye Study [21, 22].

Despite significant efforts, current treatments for advanced forms of AMD remain palliative, especially for the atrophic type [23]; therefore, prevention based on modifiable risk factors is fundamental [24, 25].

The finding that an increased risk of AMD in the Brazilian population as a whole is associated with known genetic variants has been reported in other studies, but none have evaluated the environmental risk factors for AMD in a combined analysis with these genetic factors [26, 27]. In this paper, we sought to evaluate the contributions of variants of *CFH*, *ARMS2*, and *HTRA1* genes, the most frequent environmental risk factors associated with AMD, and correlations between the two in a Brazilian population.

## Methods

### Subjects

A cross-sectional study with a control group was performed and included 236 participants 50 years of age or older, 141 of whom were individuals with AMD (the experimental group) and 95 of whom were subjects without AMD (the control group). Patients included were from the Clinical Hospital of the University of Campinas (UNICAMP) in Campinas, São Paulo, Brazil. The institution's ethics committee approved this study, and all participants provided informed consent.

### Questionnaire and clinical evaluation

All subjects were interviewed by the researcher and completed a questionnaire consisting of multiple-choice questions about demographic factors (age, gender, skin color, iris color, ocular and medical history [including cataract, hypertension, diabetes, CVD, high serum cholesterol]), family history of AMD, lifestyle (physical activity), antioxidant intake, exposure to sunlight, and smoking and drinking habits. Smoking status was assessed by whether the individual was a current smoker, exposed to environmental tobacco smoke (ETS), or had never smoked (information which was self-reported). BMI was calculated as each subject's weight in kilograms divided by height in meters squared.

### Age-related macular degeneration grading

All subjects underwent a complete ophthalmological examination, including biomicroscopy, retinography, fundoscopy, fluorescein angiography and optical coherence tomography (OCT). The study subjects were classified according to the International ARM Epidemiological Study Group definition for AMD [28]. Wet AMD was defined as the presence of at least one of the established characteristics (retinal pigment epithelial [RPE] detachment in one eye; sub-retinal or sub-RPE neovascular membranes; intraretinal, sub-retinal, or sub-epithelial pigmented scar or glial tissue; intra or sub-retinal or sub-RPE fluid or hemorrhage). Dry AMD was defined as

a geographic atrophy of the RPE with either delimited zone of partial or complete depigmentation or apparent lack of RPE, in which the choroidal vessels of the eye are visible in an area larger than 175 μm. Wet AMD and GA involving center of macula were considered to be advanced forms of AMD.

## Control group

The control group consisted of patients older than 50 years of age who presented no evidence of AMD, such as drusen or changes in the RPE. These individuals consisted of non-family companions of patients examined in the retina clinic, as well as patients examined in other subspeciality wards of the local ophthalmology department. Exclusion criteria consisted of high myopia, angioid streaks, infective or inflammatory chorioretinal disease, or trauma. All subjects received a complete ophthalmic examination, including visual acuity measurement, refraction, biomicroscopy, and fundoscopy. Fluorescein angiography was performed in patients with wet AMD, and potential subjects with suspected neovascularization from retinal angiomatous proliferation or polypoidal choroidal vasculopathy were also excluded.

## Genotyping

DNA was extracted from leukocytes present in 4 mL to 8 mL of peripheral blood. The *LOC387715 (rs10490924)*, *CFH (rs1061170)*, and *HTRA1 (rs11200638)* variants were evaluated using PCR and enzymatic digestion or PCR and direct sequencing, as in previous studies [29–31].

## Statistical analysis

The patient and control sample data were compared using Levene's test for equality of variances, and means were compared using Student's t-test. Associations between the variables were also verified using the chi-squared test and contingency tables, as well as through the calculation of the odds ratio (OR). The factors that were found to differ significantly between the two groups were considered independent variables in a stepwise discriminant analysis in which either the control group or the experimental group served as the dependent variable. To verify the relationship between risk factors and AMD severity, patients with AMD were grouped into two subgroups: patients with early AMD (mild dry form and moderate dry form) and patients with late AMD (geographic atrophy and neovascular form). The chi-squared test and Hardy–Weinberg equilibrium were employed to analyze genotypic and allelic distribution in both groups.

With the condition of having AMD (n = 141) or belonging to the control group (n = 95) serving as the dependent variable, multiple linear regression analysis was performed with the eight independent variables that, in the univariate analysis, were found to be significantly associated with the former condition. The eight independent variables were age, family history, number of *LOC387715* gene alleles, number of *CFH* gene alleles, number of *HTRA1* gene alleles, CVD, dyslipidemia, and a sedentary lifestyle. In the stepwise multiple linear regression analyses, the dependent variable was the presence or absence of AMD, and the independent variables were the genetic variants and the epidemiological risk factors. *P*-values less than 0.05 were considered statistically significant.

## Results

Two hundred and thirty-six subjects participated in this study, 141 of whom had AMD. Of the AMD cases, 70% (99/141) were classified as the advanced form of AMD in at least one eye

**Table 1. Risk factors found to be significantly associated with AMD in a comparison between AMD patients and controls.**

| Variable | AMD Cases (n = 141) | Controls (n = 95) | $\chi^2$ | P-Value | Odds Ratio (95% CI) |
|---|---|---|---|---|---|
| **Age Range (Years)** | | | 7.212 | 0.007 | 1.51 (0.88–2.58) |
| **50–75** | 67(47.5%) | 62(65.3%) | | | |
| **>75** | 74(52.5%) | 33(34.7%) | | | |
| **Family History** | | | 14.937 | <0.001 | 6.58(1.94–22.31) |
| **No** | 109(77.3%) | 91(95.8%) | | | |
| **Yes** | 29(20.6%) | 3(3.2%) | | | |
| **Unknown** | 3(2.1%) | 1(1.1%) | | | |
| **Cardiovascular Disease** | | | 6.912 | 0.009 | 2.39 (1.08–5.28) |
| **No** | 109(77.3%) | 86(90.5%) | | | |
| **Yes** | 32(22.7%) | 9(9.5%) | | | |
| **High Serum Cholesterol** | | | 3.923 | 0,048 | 1.49 (0.84–2.65) |
| **No** | 88(62.4%) | 71(74.7%) | | | |
| **Yes** | 53(37.6%) | 24(25.3%) | | | |
| **Physical Activity Level** | | | 4.632 | 0,031 | 1.39 (0.82–2.37) |
| **Good** | 69(48.9%) | 60(63.2%) | | | |
| **Low** | 72(51.1%) | 35(36.8%) | | | |

AMD = age-related macular degeneration; CI = confidence interval

(57% neovascular AMD and 13% geographic atrophy), and the remaining 42 cases (30%) were found to be not-advanced AMD.

The mean age of all participants was 73.6±7.9 years. When AMD and control groups were compared, the AMD group was significantly older (mean age 74.4±8.1 years vs. 72.24±7.4 years; OR 1.51; 95% CI: 0.88–2.58). The proportion of cases reporting a family history of AMD was significantly higher in the AMD group than in the control group (21% vs. 3%; OR 6.58; 95% CI: 1.94–22.31). Patients in the AMD group were significantly more likely to have CVD than the controls were (23% vs. 9%; OR 2.39; 95% CI: 1.08–5.28). Total cholesterol levels were significantly higher in AMD patients than in the controls (38% vs. 25%; OR 1.39; 95% CI: 0.82–2.37). Significantly more AMD patients had low physical activity levels when compared to controls (51% vs. 37%; OR 1.39; 95% CI: 0.82–2.37). Data is detailed in Table 1.

AMD was not found to be significantly associated with hypertension, diabetes, skin color, iris color, a previous cataract surgery, a smoking habit, alcohol consumption, BMI, antioxidant intake, or exposure to sunlight.

CVD was found to be significantly more frequent among advanced AMD patients than among not-advanced AMD patients (27% vs. 12%; OR: 2.29; 95% CI: 0.81–6.44).

The genotype distribution of all three variants studied differed significantly between the AMD patients and controls. The distribution and comparison of the polymorphisms between the AMD and control groups are listed in Table 2. Hardy–Weinberg equilibrium was observed in both cases and controls and for all variants: *LOC387715 (rs10490924)* in AMD patients ($p = 0.499$) and controls ($p = 0.203$); *CFH (rs1061170)* in AMD patients ($p = 0.948$) and controls ($p = 0.847$); and *HTRA1 (rs11200638)* in AMD patients ($p = 0.986$) and controls ($p = 0.515$). The frequencies of the T (rs10490924), C (rs1061170), and A (rs11200638) risk alleles were 40.0%, 46.1%, and 62.8% in patients with AMD and 23.0%, 27.3%, and 37.9% in controls, respectively.

**Table 2. Genotype distribution of rs10490924, rs1061170, and rs11200638 between AMD cases and controls.**

| Gene | AMD Patients (n = 141) | Controls (n = 95) | $\chi^2$ | P-Value | OR (95% IC) |
|---|---|---|---|---|---|
| *LOC387715* | GG–54 (38.3%) | 53 (55.8%) | 16.612 | 0.0002 | 2.21 (1.47–3.35) |
| *rs10490924* | GT–61 (43.3%) | 40 (42.1%) | | | |
| | TT–26 (18.4%) | 2 (2.1%) | | | |
| *CFH Y402H* | TT–40 (28.3%) | 49 (51.6%) | 16.841 | 0.0002 | 2.27 (1.52–3.37) |
| | TC–72 (51.1%) | 40 (42.1%) | | | |
| | CC–29 (20.6%) | 6 (6.3%) | | | |
| *HTRA1* | GG–20 (14.2%) | 34 (35.8) | 27.904 | $8.723 \times 10^{-7}$ | 2.76 (1.89–4.03) |
| *rs11200638* | GA–65 (46.1%) | 50 (52.6) | | | |
| | AA–56 (39.7%) | 11(11.6%) | | | |

T = risk allele of the *LOC387715* gene; C = risk allele of the *CFH* gene; A = risk allele of the *HTRA1* gene

**Table 3. Distribution and comparison of homozygosity of the risk allele between AMD subtypes.**

| Genotype | AMD Subtype | | P-Value |
|---|---|---|---|
| | Not-advanced | Advanced | |
| *LOC 387715(TT)* | 6 (24%) | 19 (76%) | 0.0093 |
| *CFH Y402HCC)* | 8(27.5%) | 21(72.5%) | 0.0158 |
| *HTRA1(AA)* | 15(27.2%) | 40 (72.8%) | 0.0007 |

After stratifying AMD by its subtypes, results also differed significantly between cases of not-advanced AMD and advanced AMD in the case of all three variants (Table 3) in terms of the presence of two copies of the risk allele.

Stepwise multiple linear regression analyses showed that the number of risk alleles in the *HTRA1*, *CFH*, and *LOC387715* genes were responsible for 19.7% of the AMD determination coefficient. The incremental increases in descending order of contribution to AMD development was the presence of *HTRA1* and *CFH* risk alleles, family history of AMD, *LOC387715* risk alleles, and the presence of CVD, for a total of 25.6% contribution to the AMD phenotype (Table 4). Age, dyslipidemia, and sedentary lifestyle were no longer significant when the other variables present in the table were considered.

## Discussion

Our findings confirm previous reports that factors such as age, family history of AMD, CVD, high serum cholesterol, and low physical activity are associated with an increased risk of AMD.

**Table 4. Values of the coefficient of determination and its increment in descending order of importance of the independent variable to the determination of AMD.**

| Variable | Coefficient of Determination (%) | Increase in the Coefficient of Determination (%) |
|---|---|---|
| Number of *HTRA1* alleles | 10.7% | - |
| Number of *HTRA1* and *CFH* alleles | 17.1% | 6.4% |
| Number of *HTRA1* and *CFH* alleles, family history | 21.3% | 4.2% |
| Number of *HTRA1* and *CFH* alleles, family history, number of LOC alleles | 23.9% | 2.6% |
| Number of *HTRA1* and *CFH* alleles, family history, number of *LOC* alleles, cardiovascular disease | 25.6% | 1.6% |

In a comparison of groups of individuals between 50 and 75 years of age to those 75 years of age or older, the older group was found to have a 1.5-fold increased risk of developing AMD. These results are consistent with the association between age and AMD that has been described in other studies [3, 4].

CVD and its risk factors have been significantly associated with AMD in several studies [5, 16, 26], but there are conflicting reports on this issue [32]. In a recent review, Pennington and DeAngelis concluded that there is substantial overlap among the risk factors contributing to both AMD and CVD conditions [33]. In the current study, individuals with history of CVD were found to have a 2.4-fold increase in the risk of developing AMD relative to controls, which is similar to the result obtained in the Blue Mountain Eye Study (OR: 1.57). The risk of advanced AMD was found to be double (OR: 2.29) among individuals with CVD, a finding which was consistent with the results of previous studies suggesting the possibility of a similar pathogenesis for neovascular AMD and CVD [4, 34]. The results of the current study showed that individuals with high levels of total cholesterol had a 1.5 times higher risk of AMD than controls, a finding which confirms the association between high concentrations of serum cholesterol, CVD, and stroke [16, 18]. Low physical activity, another risk factor associated with CVD, was also found to be associated with an increased risk of AMD (OR: 1.4). These findings are consistent with other reports suggesting a protective effect of routine exercise [18, 20, 35]. CVD, which may share pathogenic pathways with AMD, is another example of a complex condition that has been shown to be a significant risk factor for AMD in general and for advanced forms of AMD in particular. Thus, potentially modifiable lifestyle factors may be a useful focus of public health awareness campaigns. Projects prioritizing blindness prevention in the elderly should consider similar CVD prevention programs currently available in Brazilian health care system.

The frequency of a family history of AMD found in this study (OR: 6.6) was lower when compared to the Beaver Dam Eye Study (OR: 10.3) and the Rotterdam Eye Study (OR: 14.3). The frequency, however, was still significant, confirming the importance of family history of AMD that has been described in previous reports [5]. Our results on the influence of the *LOC387715*, *CFH Y402H*, and *HTRA1* variants are consistent with the literature: the individuals with risk alleles were almost twice as likely to be part of the AMD group relative to the control group (OR: 2.21, 2.27 and 2.76 respectively) [9, 36]. The results of studies on the *CFH Y402H* variant in another Brazilian population were similar to those found herein [26, 27]. Our results on the *HTRA1* gene were similar to those of several publications that have reported the *rs11200638* variant to be the one most commonly associated with susceptibility to AMD; in the literature, this variant has also consistently been reported to be a stronger risk factor for wet AMD than for dry AMD [9]. When comparing the frequency of the candidate variants from *CFH*, *LOC387715/ARMS2* and *HTRA1* between advanced AMD and not-advanced AMD, there was a strong association with the cumulative presence of the risk alleles, results confirmed by other studies [9, 37]. The risk factors in a combined analysis showed that genetic factors remained significantly contributing up to 19% to AMD development. Among them, the risk variant in the *HTRA1* gene was the major contributor in agreement to what was reported in other populations. The result of this analysis allows us to conclude that the combined weight of the variables as number of *HTRA1* and *CFH* alleles, family history, number of LOC alleles, CVD, in the determination of AMD, is 25.6%, a result with statistical significance. It also indicates that the investigation of this theme deserves attention, because other variables not considered in this study certainly play a relevant role for the occurrence of this condition.

Unlike previous published reports, the current study failed to find a statistically significant association between AMD and frequently reported risk factors such as smoking [14, 17], cataracts, and/or a prior cataract surgery [22]. Cross-sectional case-control studies are, by nature

of their design, subject to several inherent limitations, and the present study is no exception. All subjects (both the AMD patients and the controls) were recruited from hospital clinics, a factor which has the potential to introduce bias, and they were not individually matched, though all analyses were adjusted for age and gender.

Despite its small sample size, this study confirms the association between AMD and risk factors such as ageing, a family history of AMD, and cardiovascular disease, and the results for this Brazilian population were largely similar to results found on other populations worldwide. Because AMD is a complex disease, the *CFH*, *LOC387715/ARMS2*, and *HTRA1* variants were also found to be a significant risk factor for AMD development in this Brazilian population and are likely to be involved in the etiology of AMD. In addition, the presence of homozygosity of the risk alleles was found to increase susceptibility to advanced forms of the disease.

Determining the status of Brazilian patients with this heterogeneous disease and inherent peculiar behavioral, clinical, and genetic risk factors will contribute to guide clinical management decisions and impact on AMD outcomes.

Larger studies with a greater number of patients from different regions of Brazil, will help to increase our knowledge on the risk of AMD development in this highly admixed population.

## Supporting information

**S1 Fig. Patients environmental risk factors data chart.**
(DOCX)

**S2 Fig. Controls environmental risk factors data chart.**
(DOCX)

**S3 Fig. Patients molecular genetics data chart.**
(DOCX)

**S4 Fig. Controls molecular genetics data chart.**
(DOCX)

**S1 File.**
(DOCX)

**S2 File.**
(PDF)

## Author Contributions

**Conceptualization:** Priscila H. H. Rim, José Paulo C. de Vasconcellos, Antonia P. Marques-de-Faria.

**Data curation:** Priscila H. H. Rim, Flavio M. C. Medina, Daniela P. D. Sacconi, Tamires P. Lana, Fabio E. Hirata.

**Formal analysis:** Fabio E. Hirata, Luis A. Magna, Antonia P. Marques-de-Faria.

**Funding acquisition:** Mônica B. de Melo.

**Investigation:** Priscila H. H. Rim, Fabio E. Hirata, Antonia P. Marques-de-Faria.

**Methodology:** Priscila H. H. Rim, Mônica B. de Melo, Daniela P. D. Sacconi, Tamires P. Lana, Luis A. Magna, Antonia P. Marques-de-Faria.

**Project administration:** Priscila H. H. Rim.

**Supervision:** José Paulo C. de Vasconcellos, Mônica B. de Melo.

**Writing – original draft:** Priscila H. H. Rim, Luis A. Magna, Antonia P. Marques-de-Faria.

**Writing – review & editing:** Priscila H. H. Rim, José Paulo C. de Vasconcellos, Mônica B. de Melo, Flavio M. C. Medina, Antonia P. Marques-de-Faria.

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
