## [Decision Letter · Decision Letter 0]

24 Jan 2022

PONE-D-21-33908CORRELATION BETWEEN GENETIC AND ENVIRONMENTAL RISK FACTORS FOR AGE-RELATED MACULAR DEGENERATION IN BRAZILIAN PATIENTSPLOS ONE

Dear Dr. RIM,

Thank you for submitting your manuscript to PLOS ONE. After careful consideration, we feel that it has merit but does not fully meet PLOS ONE’s publication criteria as it currently stands. Therefore, we invite you to submit a revised version of the manuscript that addresses the points raised during the review process.

The expert reviewer has raised questions about your classification of AMD cases, about the questionnaire that you used, and about the diversity of the Brazilian population. Please address these concerns. Although importance is not a criterion for publication, it would be helpful if you could address the novelty of your research.

We look forward to receiving your revised manuscript.

Kind regards,

Alfred S Lewin, Ph.D.

Academic Editor

PLOS ONE

Journal Requirements:

2. Please ensure that you have specified (1) whether consent was informed, (2) what type you obtained (for instance, written or verbal, and if verbal, how it was documented and witnessed). If your study included minors, state whether you obtained consent from parents or guardians. If the need for consent was waived by the ethics committee and (3) If you are reporting a retrospective study of medical records or archived samples, please ensure that you have discussed whether all data were fully anonymized before you accessed them and/or whether the IRB or ethics committee waived the requirement for informed consent. If patients provided informed written consent to have data from their medical records used in research, please include this information.

Reviewers' comments:

Reviewer's Responses to Questions

**Comments to the Author**

1. Is the manuscript technically sound, and do the data support the conclusions?

Reviewer #1: Yes

2. Has the statistical analysis been performed appropriately and rigorously? 

Reviewer #1: Yes

3. Have the authors made all data underlying the findings in their manuscript fully available?

Reviewer #1: Yes

4. Is the manuscript presented in an intelligible fashion and written in standard English?

Reviewer #1: Yes

5. Review Comments to the Author

Reviewer #1: I believe the study is well performed and clearly written. However, there are a few issues that called my attention.

1. The classification and definition of AMD shoud be reviewed. Authors classisfy the disease in early and advance, (early including early and intermediate AMD). I believe this is not accurate and can lead to confussion, it would be better to classify patients into early, intermediate and advanced AMD. If this is not possible, then at least the term early should be changed to "not-advanced" AMD.

Further, authors should specify if the clinical features listed as diagnostic for wAMD and geographic atrophy were based on clinical fundoscopy, fundus photography, OCT or other methods. Moreover, some of the clinical characteristics listed to define wAMD are not specific for the disease (RPE detachment, epiretinal membrane, hard exudates, phocoagulation scars) while others are missing (intraretinal or subretinal fluid). The same is true for the definition of geographic atrophy (Ex: hypopigmentation does not imply the presence of atrophy). Why is an area of 175microns used to determine atrophy? A deep review to the inclusion and exlcusion criteria should be done and better explained.

2. It would be very helpful to know which questionaire was used to assess demographic and general health characteristics. Is it a validated questionaire, is it one proposed by the authors (if so, further ditails should be incuded).

3. Brazil has a very diverse population, something very important when performing genetic studies. How was this addressed? We patients from different ethnicities included?

4.Methods sections would be clearer if devided in subheadings

5.It is not clear the novelty offered by the results of this manuscript? what makes it special or different from previous published works? This should be explained in the discussion sections, otherwise, it just seems like a confirmation of things which are already known.

6. PLOS authors have the option to publish the peer review history of their article (what does this mean?). If published, this will include your full peer review and any attached files.

Reviewer #1: No

---

## [Author Response · Author response to Decision Letter 0]

18 Apr 2022

Dear Dr Lewin and reviewer,

Thank you for considering our manuscript for publication. We hope that after revision it meets criteria and merit. 

The expert reviewer has raised relevant questions that will enhance this study, we really appreciate.

kind regards,

Priscila Rim

---

## [Editor Report · Decision Letter 1]

9 May 2022

CORRELATION BETWEEN GENETIC AND ENVIRONMENTAL RISK FACTORS FOR AGE-RELATED MACULAR DEGENERATION IN BRAZILIAN PATIENTS

PONE-D-21-33908R1

Dear Dr. RIM,

We’re pleased to inform you that your manuscript has been judged scientifically suitable for publication and will be formally accepted for publication once it meets all outstanding technical requirements.

Kind regards,

Alfred S Lewin, Ph.D.

Section Editor

PLOS ONE
---

## [Editor Report · Acceptance letter]

23 May 2022

PONE-D-21-33908R1 

Correlation between genetic and environmental risk factors for age-related macular degeneration in Brazilian patients 

Dear Dr. RIM:

I'm pleased to inform you that your manuscript has been deemed suitable for publication in PLOS ONE. Congratulations! Your manuscript is now with our production department. 

Kind regards, 

on behalf of

Dr. Alfred S Lewin 

Section Editor

PLOS ONE